# In Vitro Activity of Eravacycline against Carbapenemase-Producing Gram-Negative Bacilli Clinical Isolates in Central Poland

**DOI:** 10.3390/biomedicines11071784

**Published:** 2023-06-21

**Authors:** Małgorzata Brauncajs, Filip Bielec, Anna Macieja, Dorota Pastuszak-Lewandoska

**Affiliations:** 1Department of Microbiology and Laboratory Medical Immunology, Medical University of Lodz, 90-151 Lodz, Poland; malgorzata.brauncajs@umed.lodz.pl (M.B.); dorota.pastuszak-lewandoska@umed.lodz.pl (D.P.-L.); 2Medical Microbiology Laboratory, Central Teaching Hospital of Medical University of Lodz, 92-213 Lodz, Poland

**Keywords:** eravacycline, antimicrobial susceptibility, Gram-negative rods, carbapenem resistance

## Abstract

Eravacycline is a novel antibiotic of the tetracycline class with activity against a broad spectrum of clinically significant bacteria, including multi-drug-resistant organisms. For this reason, it may be an alternative to treating critical infections of this etiology. We aimed to assess the in vitro effectiveness of eravacycline to carbapenemase-producing Gram-negative bacilli clinical isolates identified in hospitals in Łódź, Poland. We analyzed 102 strains producing KPC, MBL, OXA-48, GES, and other carbapenemases. Eravacycline susceptibility was determined following the EUCAST guidelines. The highest susceptibility was found in KPC (73%) and MBL (59%) strains. Our results confirmed in vitro the efficacy of this drug against carbapenem-resistant strains. However, eravacycline has been indicated only for treating complicated intra-abdominal infections, significantly limiting its use. This aspect should be further explored to expand the indications for using eravacycline supported by evidence-based medicine. Eravacycline is one of the drugs that could play a role in reducing the spread of multidrug-resistant microorganisms.

## 1. Introduction

Eravacycline is a fully synthetic fluorocycline antibiotic of the tetracycline class with activity against clinically significant Gram-negative, Gram-positive aerobic, and anaerobes, except for *Pseudomonas aeruginosa*. This includes most bacteria that are resistant to cephalosporins, fluoroquinolones, β-lactam/β-lactamase inhibitors, multidrug-resistant strains, and carbapenem-resistant Enterobacterales strains, and the majority of anaerobic pathogens [1].

Eravacycline has been indicated for the treatment of complicated intra-abdominal infections in adults. This antibiotic, similar to other tetracyclines, exhibits a typically bacteriostatic activity; in addition, eravacycline also has bactericidal activity against some strains of *Acinetobacter baumannii*, *Escherichia coli*, and *Klebsiella pneumoniae* in vitro [1,2]. Erevacycline’s chemical structure and mechanism of action are presented in Figure 1.

Currently, as for Gram-negative bacilli, EUCAST gives breakpoints only for *E. coli*. With regard to *Acinetobacter* spp., the evidence is insufficient, and no breakpoints were awarded to *Pseudomonas* spp. [3]. Table 1 summarizes the current eravacycline MIC breakpoints for various bacteria.

The aim of this study was to assess the in vitro effectiveness of eravacycline on carbapenemase-producing Gram-negative bacilli clinical isolates identified in hospitals in Łódź, Poland.

## 2. Materials and Methods

In total, 102 strains producing KPC, MBL, OXA-48, GES, and other unidentified carbapenemases were investigated. All strains were isolated from the clinical samples: bronchial alveolar lavage (BAL), blood, urine, rectal swab (screening for carbapenemase-producing organisms, CPO), lower respiratory specimen (other than BAL), intraoperative swab, nasal swab, wound swab, and pressure ulcer swab.

All bacteria were stored in Viabank^TM^ storage beads (Medical Wire and Equipment, Corsham, UK) at a maximum of −80 °C for six months and were regenerated on Columbia Agar with 5% sheep blood (Thermo Fisher Scientific, Waltham, MA, USA) for 18–24 h at 37 °C. The isolates were tested for their susceptibility to meropenem-vaborbactam, imipenem-cilastin-relebactam, and ceftazidime-avibactam using MIC (minimum inhibitory concentration) test strips (Liofilchem, Roseto degli Abruzzi, Italy) and the same standardized inoculum. Drug susceptibility was determined on a standard Mueller-Hinton Agar (Thermo Fisher Scientific, Waltham, MA, USA) and incubated for 18 ± 2 h at 35 ± 1 °C following the European Committee on Antimicrobial Susceptibility Testing (EUCAST) guidelines [3]. The ability of all studied strains to produce carbapenemases was assessed and confirmed as described previously [4].

Descriptive statistics were prepared using Microsoft Excel 2019 software (Microsoft Corporation, Redmond, Washington, DC, USA).

## 3. Results

The tested strains came from the same collection described in the authors’ previous publication [4]. The group of tested bacteria consisted of 50 *K. pneumoniae*, 7 *E. coli*, 15 *P. aeruginosa*, and 26 *A. baumannii*. The remaining four isolates were single strains of *Aeromonas sobria*, *Klebsiella varicola*, *Psudomonas alcaligenes*, and *Pseudomonas putida*. The distribution of the detected resistance mechanisms is presented in Table 2.

Among Gram-negative Enterobacterales, 86% of *E. coli* and 66% of *K. pneumoniae* showed susceptibility to eravacycline (with a breakpoint at 0.5 mg/L given for *E. coli* only, though we extrapolated it). When analyzing isolates of non-fermenting Gram-negative bacilli, 20% of *P. aeruginosa* and 50% of *A. baumannii* strains had a MIC above 0.5 and showed susceptibility to this drug. EUCAST did not define a breakpoint for these species and even for *Pseudomonas* spp. Susceptibility testing is not recommended. However, reference to the existing breakpoints for other Gram-negative bacilli allowed us to compare our in vitro susceptibility results with the data available in the literature.

When analyzing the resistance mechanisms to carbapenems, the highest susceptibility was found in KPC strains: 73%, MBL strains: 59%, and among OXA-48 strains: 40% of isolates showed susceptibility. The MIC_50_ value for these strains was 0.38 mg/L, and the MIC_90_ value was 1 mg/L.

Table 2 presents the values of the growth inhibition zone range, MIC range, MIC_50_, and MIC_90_ for eravacycline. Figure 2 and Figure 3 present the antimicrobial in vitro activity of eravacycline against the species analyzed in this study.

## 4. Discussion

In our study, we analyzed the in vitro effectiveness of eravacycline to carbapenemase-producing Gram-negative bacilli clinical isolates identified in hospitals in Łódź, Poland. Currently, eravacycline is used to treat complicated intra-abdominal infections, which are the second largest type of infection in the intensive care unit (ICU) and the second leading cause of infection-related mortality in ICUs. What is alarming is that more and more often, they are caused by multidrug-resistant strains.

Connors et al. [5] compared eravacycline with the drug from the group of glycylcycline antibiotics, namely tigecycline, and found that eravacycline had better pharmacokinetics than tigecycline and achieved higher serum concentrations with excellent penetration into the epithelial lining fluid and alveolar macrophages. In phase III clinical trial, eravacycline was used at a dose of 1 mg/kg body weight administered intravenously (IV) every 12 h to treat complicated intra-abdominal infections and appeared to be non-inferior to ertapenem in terms of effectiveness and safety of use [1]. These results were also confirmed in other clinical studies [6,7,8].

Furthermore, in a complicated infection of the urinary tract, eravacycline (1.5 mg/kg IV every 24 h, tapered to 200 mg orally every 12 h from day 3) was also not inferior to levofloxacin [1]. In other clinical trials, eravacycline has been shown with a favorable safety profile, and no dose adjustment was required in patients with renal impairment. The drug could also be safely administered to patients allergic to penicillin. [9]. Such results could suggest the possibility of extending the recommendations included in the Summary of Product Characteristics (SmPC).

An analysis of fluoroquinolones revealed that they were characterized by growing resistance and demonstrated a significant deleterious impact on gastrointestinal microbiota. Hence, they should be used in a limited way because uncontrolled and often repeated therapeutic cycles with these drugs in the treatment of urinary tract infections are a selection factor for drug-resistant mutants [10,11,12]. In this situation, eravacycline was a better choice for treating complicated urinary tract infections. However, it is worth noting once again that such an indication is not found in SmPC: eravacycline is not indicated but only “considered” in the treatment of complicated urinary tract infections, as clinical trials [13] showed no efficacy in the combined endpoints of clinical treatment and microbiological success. However, Grossman et al. [14] confirmed the in vitro activity of eravacycline in preformed biofilms formed by uropathogenic *E. coli* isolates. We also evaluated eravacycline’s susceptibility to bacterial strains isolated, i.a., from urine samples, and our results revealed that 86% of *E. coli* strains were susceptible to eravacycline. Additionally, the study performed by Zou et al. [15] reported the susceptibility of carbapenem-resistant *E. coli* isolates to eravacycline (MIC_50_ and MIC_90_ were 0.25 mg/L and 0.5 mg/L, respectively) and showed that as many as 92% of *E. coli* isolates were susceptible. An even higher percentage of *E. coli* were found to be susceptible to eravacycline, as revealed in another study [16].

Generally, the results of several studies clearly indicated the in vitro activity of eravacycline against carbapenem-resistant Gram-negative bacteria. Abdallah et al. [17] demonstrated the in vitro efficacy of eravacycline against CPO-positive and carbapenem-insusceptible isolates of *A. baumannii*: the eravacycline MIC values ranged from 0.06 to 4 mg/L. In our study, half of *A. baumannii* strains showed susceptibility to this drug (MIC below 0.5 mg/L). The research performed by Sutcliffe et al. [15] revealed that 52 isolates of *A. baumannii* had a MIC ≤ 0.016–4 mg/L and 145 isolates of *P. aeruginosa* had a MIC 1–32 mg/L. In our study, 20% of *P. aeruginosa* strains were susceptible to eravacycline. Different results were reported by Morrisey et al. [16], who found that only 1% of isolates of *P. aeruginosa* were susceptible to eravacycline and as many as 70.5% of strains of *A. baumannii*.

In the research of Zou et al. [15], the susceptibility of *K. pneumoniae* to eravacycline was shown in 53.1% of cases. In our study, the percentage of *K. pneumoniae* susceptible to this tetracycline was a little higher, reaching 66%, but not as high as that reported by Morrissey et al. [16], who demonstrated the antibacterial activity of eravacycline against 90.6% of these Gram-negative rod strains.

Although efficacy in OXA-48-like producers has not been specifically evaluated, eravacycline is still a potential candidate for treating the nosocomial infections caused by these pathogens [18]. Our study showed that 40% of OXA-48-positive strains were susceptible to eravacycline, as the MIC value was below 0.5 mg/L.

We also found that 59% of MBL-producing strains and 73% of KPC-positive isolates were susceptible to eravacycline, with MIC values below 0.5 mg/L. This clearly indicates that this drug could be applied if there are hardly any alternatives for the treatment of infections of a multi-drug-resistant etiology. This is in line with the results obtained by others who demonstrated the most potent inhibitory concentrations for eravacycline when compared with other antibiotics that were tested against a set of multidrug-resistant CRE isolates [15,19]. The legitimacy of using the new tetracycline in treating infections caused by carbapenemase-producing strains was also confirmed by Huges et al. [20]. However, they noted that a lack of clinical data precluded the current recommendations for the implementation of eravacycline in the treatment of invasive CPO infections when other established therapies are available.

The interesting finding in our study is the evidence of the relationship of MIC values to the specific mechanism of acquired resistance to carbapenems—see Table 2. The results show that while the MIC_50_ values were similar, the MIC_90_ values differed—they were at 1–3 mg/L for KPC, GES, and OXA-48 and as much as four times higher for the MBL and other CIM-positive strains. MBLs are encoded by the so-called gene cassettes located within integrons, i.e., genetic elements capable of “catching” and accumulating cassettes and their expression. Sometimes even multi-resistance [21,22]. MBL is not inhibited to any extent by β-lactam inhibitors (clavulanate, sulbactam, and tazobactam). Among the most recently introduced new fifth-generation cephalosporins, only cefiderocol exhibited in vitro activity against MBL-positive strains. Recently approved medicines containing new β-lactamase inhibitors ceftazidime/avibactam, imipenem/relebactam, and meropenem/vaborbactam were found to have no activity against these organisms [23,24]. Perhaps these characteristics are the reason for the high MIC values.

## 5. Conclusions

Eravacycline is an antibacterial drug of the tetracycline group with in vitro activity against carbapenem-resistant bacteria. For this reason, it may be an alternative when treating infections of this etiology. Our results confirmed in vitro the efficacy of this drug against CPO-positive strains.

However, eravacycline has been indicated only for the treatment of complicated intra-abdominal infections, which significantly limits its use. Currently, in severe CPO-positive infections, high-dose strategies and synergies involving new combinations of old β-lactams with new β-lactamase inhibitors should be considered as they may contribute to therapeutic success, e.g., the synergy of ceftazidime/avibactam with aztreonam or meropenem/vaborbactam with aztreonam.

The identification of the resistance phenotype and implementation of methods that can significantly accelerate the assessment of susceptibility to antibiotics are highly essential as they could help reduce the spread of multidrug-resistant microorganisms. Eravacycline is one of the drugs that potentially could play a role in this process.

## Figures and Tables

**Figure 1 biomedicines-11-01784-f001:**
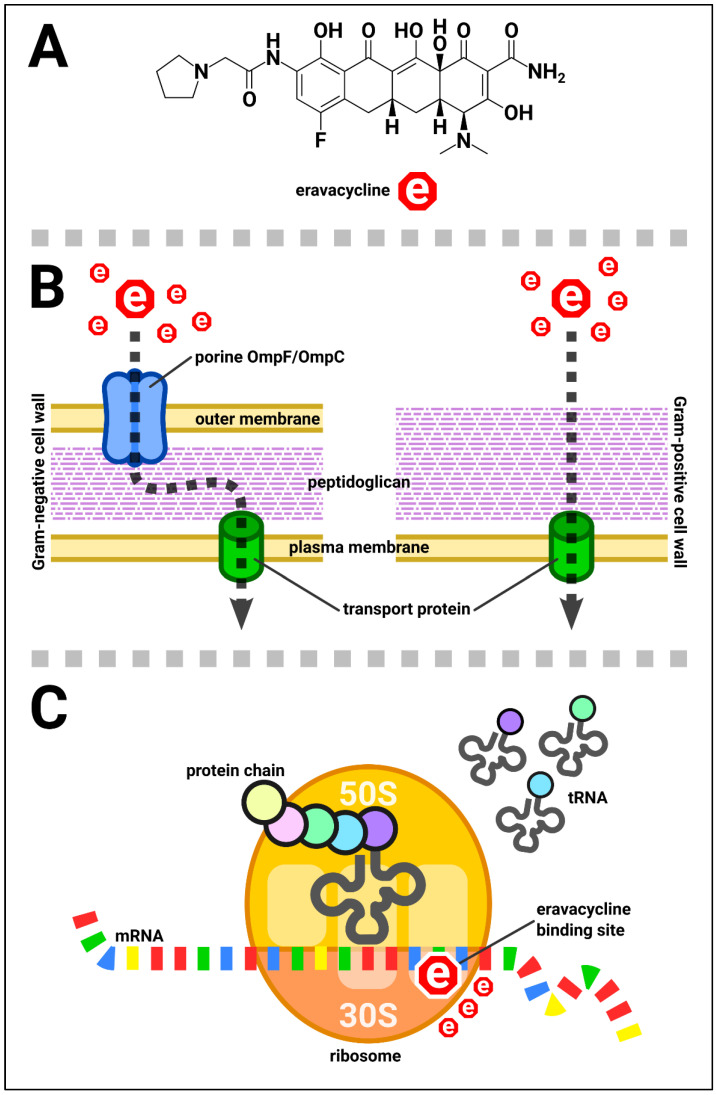
Eravacycline: (**A**)—Chemical structure; (**B**)—Transport through the bacterial cell wall; (**C**)—Mechanism of antimicrobial action: eravacycline inhibits protein synthesis by binding to the 30S subunit of the bacterial ribosome, preventing the amino-acyl tRNA from binding to the acceptor site on the mRNA–ribosome complex.

**Figure 2 biomedicines-11-01784-f002:**
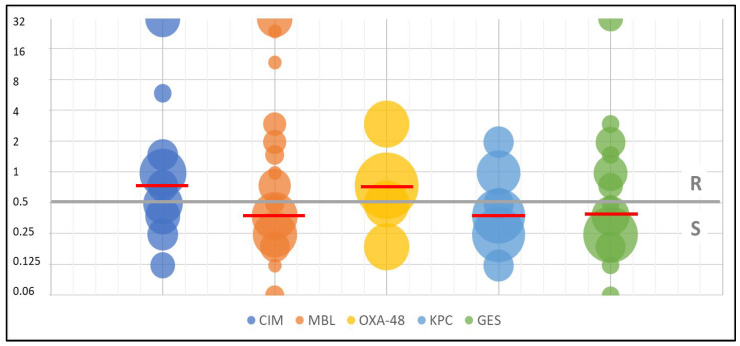
The antimicrobial in vitro activity of eravacycline against carbapenemase-producing bacteria depending on a resistance mechanism—MIC test strip method. The size of the bubble depends on the percentage of strains with a given MIC value, the grey line indicates the breakpoint between susceptible and resistant, and the red lines indicates the average MIC values (KPC—*Klebsiella pneumoniae* carbapenemase; OXA-48—oxacillinase-48; GES—Guiana extended-spectrum; MBL—metallo-β-lactamase; CIM—carbapenem inactivation method; R—resistant; S—susceptible).

**Figure 3 biomedicines-11-01784-f003:**
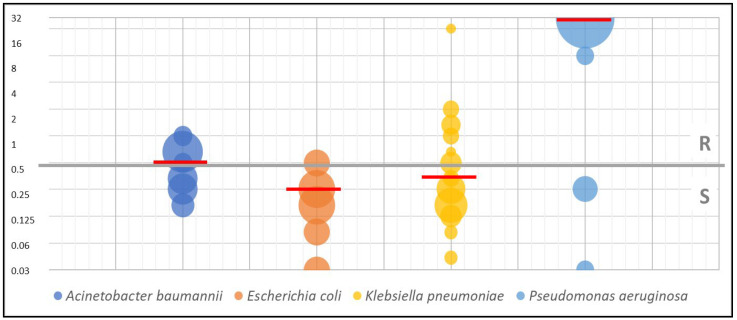
The antimicrobial in vitro activity of eravacycline against carbapenemase-producing bacteria depending on species—MIC test strip method. The bubble size depends on the percentage of strains with a given MIC value, the grey line indicates the breakpoint between susceptible and resistant, and the red lines indicate the average MIC values (R—resistant; S—susceptible).

**Table 1 biomedicines-11-01784-t001:** Summary of MIC breakpoints for eravacycline. Adapted from [3] 2023 EUCAST.

Organism/Organisms’ Group Name	Breakpoint (S≤; R>) [mg/L]
*Escherichia coli*	0.5
*Staphylococcus aureus*	0.25
*Enterococcus* spp.; Viridans group streptococci	0.125
other Enterobacterales; other *Staphylococcus* spp.; other *Streptococcus* spp.; *Acinetobacter* spp.; *Haemophilus influenzae*; *Neisseria* spp.; *Moraxella catarrhalis*	Insufficient evidence that the organism or group is a good target for therapy with the agent.
*Pseudomonas* spp.	No breakpoints. Susceptibility testing is not recommended.

**Table 2 biomedicines-11-01784-t002:** The antimicrobial in vitro activity of eravacycline against carbapenemase-producing species.

Resistance Mechanism	N	MIC_50_ [mg/L]	MIC_90_ [mg/L]	MIC Range [mg/L]
CIM	26	0.75	32	0.047→32
MBL	58	0.38	32	0.047→32
OXA-48	6	0.5	3	0.19–3
KPC	11	0.38	1	0.125–2
GES	35	0.38	2	0.047→32

MIC—minimum inhibitory concentration; MIC_50_—MIC required to inhibit the growth of 50% of bacteria; MIC_90_—MIC required to inhibit the growth of 90% of bacteria; KPC—Klebsiella pneumoniae carbapenemase; OXA-48—oxacillinase-48; GES—Guiana extended-spectrum; MBL—metallo-β-lactamase; CIM—carbapenem inactivation method.

## Data Availability

Not applicable.

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
