# Peer review of "In Vitro Activity of Eravacycline against Carbapenemase-Producing Gram-Negative Bacilli Clinical Isolates in Central Poland"

_biomedicines, 2023, doi:10.3390/biomedicines11071784_

Round 1

Reviewer 1 Report

I have reviewed the manuscript entitled "In vitro activity of eravacycline against carbapenemase-producing Gram-negative bacilli clinical isolates in central Poland" by a Brauncajs et al.

The authors tested sensitivity testing for 102 strains producing carbapenemases against  eravacycline which belongs to tetracycline group. The strains were also tested for susceptibility to meropenem-vaborbactam, imipenem- cilastin-relebactam, and ceftazidime-avibactam using E test strips. The strains are E. coli, K. pneumoniae, Ps. aeruginosa and A. baumannii. 

This work is very short research includes only the methodology of susceptibility testing against some antibiotics. 

The  comments

Figure 1 is not important because the mechanism is mentioned in the text

Why did the authors test another tetracyclines such as doxycyclines, tetracycline, oxytetracycline, minocycline, tigecycline to compare the results with eravacycline.

The authors tested Pseudomonas against eravacycline however they mentioned in the literature lines 26-28 "Eravacycline is a fully synthetic fluorocycline antibiotic of the tetracycline class with activity against clinically significant gram-negative, gram-positive aerobic, and anaerobes, except for Pseudomonas aeruginosa.

What is the number of E. coli, K. pneumonaie, P. aerugionsa and A. bumannii strains and what is the type of carbapenmases

The authors' results showed that there is relation between the type of carabapenemases and the MIC of Eravacycline. Could you explain this finding?

Table 1 is not clear

Author Response

Thank you for your comments and suggestions regarding our manuscript. The corrections introduced on the reviewer's advice will improve the article's quality and its better reception and understanding by readers.
The 'Introduction' has been reformulated to not duplicate the information in Figure 1.
In the 'Results' section, information on the number of individual species and the purpose of determination of AST for Pseudomonas has been added.
The 'Discussion' was extended to include an analysis of the relationship between the types of carbapenemases and the obtained MIC values.
In our study, we did not see AST for other tetracyclines, mainly for financial reasons. We wanted to focus on eravacycline as a newly available antibiotic that has the potential to treat infections with CPO strains.

Reviewer 2 Report

The authors investigated the MIC of eravacycline against carbapenemase-producing gram-negative bacilli and concluded that eravacycline is one of the drugs that could play a role in reducing the spreading of multidrug-resistant microorganisms. This study is significant for the clinical settings. However, definition of the MIC of eravacycline is wrong, which the authors defined the MIC of 0.5 against all bacteria. According to the EUCAST, the MIC breakpoints of eravacycline for Enterococcus spp. and Enterobacterales were 0.125 and 0.5, respectively. Thus, the authors can not mention the susceptibility. In the other words, they can mention fact on the MIC of eravacycline against various bacteria. Moreover, the authors need to add a new Table regarding the MIC of eravacycline of the EUCAST. Finally, in the section of discussion, the authors should describe the content of references of 5 and 1 in detail: for the reference of 5, what does mean a simpler pharmacokinetics (L109-111)? For the reference of 1, what is non-inferior in eravacycline to in other antibiotics?

Author Response

Thank you for your comments and suggestions regarding our manuscript. The corrections introduced on the reviewer's advice will improve the article's quality and its better reception and understanding by readers.
The introduction has been reformulated to include information on all current breakpoints for eravacycline.
The 'Discussion' section explains the passages that the reviewer pointed out.

Round 2

Reviewer 1 Report

The authors did the most of the comments 

Reviewer 2 Report

All were revised appropriately.